# Automated Detection of Causal Inference Opportunities: Regression Discontinuity Subgroup Discovery

**Tony Liu**                                                        *liutony@seas.upenn.edu*
*University of Pennsylvania, Roblox*

**Patrick Lawlor**                                                  *lawlorp1@chop.edu*
*Children's Hospital of Philadelphia*

**Lyle Ungar**                                                      *ungar@cis.upenn.edu*
*University of Pennsylvania*

**Konrad Kording**                                                  *koerding@gmail.com*
*University of Pennsylvania*

**Rahul Ladhania**                                                  *ladhania@umich.edu*
*University of Michigan*

**Reviewed on OpenReview:** *https://openreview.net/forum?id=cdRYoTyHZh*

## Abstract

The gold standard for the identification of causal effects are randomized controlled trials (RCT), but RCTs may not always be feasible to conduct. When treatments depend on a threshold however, such as the blood sugar threshold for diabetes diagnosis, we can still sometimes estimate causal effects with regression discontinuities (RDs). RDs are valid when units just above and below the threshold have the same distribution of covariates and thus no confounding in the presence of noise, establishing an as-if randomization. In practice however, implementing RD studies can be difficult as identifying treatment thresholds require considerable domain expertise – furthermore, the thresholds may differ across subgroups (e.g., the blood sugar threshold for diabetes may differ across demographics), and ignoring these differences can lower statistical power. Finding the thresholds and to whom they apply is an important problem currently solved manually by domain experts, and data-driven approaches are needed when domain expertise is not sufficient. Here, we introduce Regression Discontinuity SubGroup Discovery (RDSGD), a machine-learning method that identifies statistically powerful and interpretable subgroups for RD thresholds. Using a medical claims dataset with over 60 million patients, we apply RDSGD to multiple clinical contexts and identify subgroups with increased compliance to treatment assignment thresholds. As treatment thresholds matter for many diseases and policy decisions, RDSGD can be a powerful tool for discovering new avenues for causal estimation.

# 1   Introduction

Many questions in data science are ultimately causal in nature, yet evaluating causal questions through experimental randomization can be costly or otherwise infeasible (Musci and Stuart, 2019). There are numerous methods that estimate causality from observational data, but many rely on the key assumption of *no unobserved confounding*, which is generally difficult to justify in realistic data settings (Hernán and Robins, 2020). However, econometricians have been developing study designs that can make credible causal claims from observational data (Leamer, 1983; Angrist and Pischke, 2010). These study designs address confounding by exploiting naturally occurring randomness in the data, so-called *quasi-experiments* (Angrist and Pischke, 2008; Liu et al., 2021).

We focus on the regression discontinuity (RD), a specific quasi-experimental method for evaluating causal effects from observational data where a cutoff in an observed continuous *running variable* determines treatment assignment  (Hahn et al., 2001). Such a situation may arise when treatment depends on a threshold. For example, when a patient's blood sugar level (measured by A1C %) is above 6.5%, they are diagnosed as diabetic (American Diabetes Association, 2010) and hence eligible for treatment assignment. Critically, RDs are more robust to confounding than other observational causal inference methods (Lee and Lemieux, 2009), as the cutoff in treatment assignment provides "as-if" randomization for individuals just above and just below the cutoff: a patient with an A1C of 6.5%, on average, is not materially different from a patient with an A1C of 6.4%, yet the former is diagnosed with diabetes and treated for the disease while the latter is not.  If other covariates smoothly vary with the running variable, then the "just below" subjects arguably have the same (observed and unobserved) covariate distribution as the "just above" subjects. Because of this "as-if" randomization, RDs allow us to estimate treatment effects at the threshold without explicit randomization. RD opportunities are present in education (Valentine et al., 2017), political science (Skovron and Titiunik, 2017), criminal justice (Berk and Rauma, 1983), and are particularly natural in medicine, where thresholds govern decisions in many diseases, e.g. diabetes, coronary artery disease, and cancer (Petersen et al., 2020; Scott et al., 2022; Oeffinger et al., 2015).

Despite the ubiquity of such RD opportunities, RDs are underutilized (Moscoe et al., 2015; Marinescu et al., 2018). Because *a priori* knowledge of the treatment threshold is needed to use an RD, the typical study design approach is a "top-down" process, in which a domain expert hypothesizes that a particular data-generating process might yield an RD, followed by verification of study validity by examining the data. Often enough, the RD opportunity is underpowered due to sample size limitations (Naidech et al., 2020; McKenzie, 2016). Identifying potential RDs is an ad-hoc process that relies heavily on human intuition and domain expertise, and thus does not scale well to the vast amounts of high-dimensional data we have available today.

This holds especially true as treatment thresholds in practice are often multi-faceted, with heterogeneity in *treatment assignment* as a function of other covariates. For example, in medicine, though diagnostic criteria for diabetes ostensibly are made only according to blood sugar levels, the risk for the disease varies by gender, race, and age categories, leading to different clinical decisions where official guidelines may not always be followed. Ignoring these differences can lower statistical power. As treatment assignment thresholds become more complex, it becomes difficult for domain experts to generate study designs and verify them. Given the many domains where RDDs are a useful tool for causal inference, a "bottom-up" data-driven approach would streamline and scale RD study discovery, unlocking opportunities for more interpretable and well-powered causal studies.

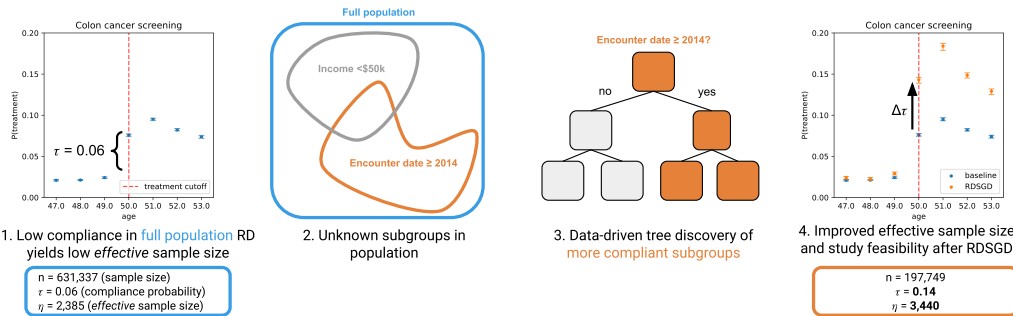

Figure 1: **This work develops RDSGD, a machine learning method for discovering RD opportunities that improve study feasibility by searching for subgroups with higher TAU compliance.** We show our colon cancer case study (Section 6) as an example.

Here we propose a data-driven method, Regression Discontinuity Subgroup Discovery (RDSGD), to learn RD subgroups with different treatment assignment thresholds (see Figure 1 for an illustration of our approach). We frame regression discontinuity discovery similarly to the task of conditional average treatment effect (CATE) estimation (Section 3). Note that our method differs from CATE estimation by focusing on heterogeneity in treatment *assignment* rather than in treatment *effects*. We introduce a novel framework targeting higher *effective sample sizes* of the discovered subgroups to maximize statistical power and maintain interpretability (Section 4). We show the utility of our approach through both synthetic experiments (Section 5) and a case study using a medical claims dataset consisting of over 60 million patients (Section 6). We apply our method to three clinical contexts and discover subgroups for each setting, with some that are validated by domain knowledge and others that show promise for potential studies. RDSGD can not only discover new opportunities for quasi-experimental studies but also provide actionable interpretability in the form of treatment assignment subgroups, which can be easily understood by practitioners (Section 7).

## 2 Related Work

Automatic regression discontinuity discovery has been explored in related contexts. Porter and Yu (2015) propose a statistical testing framework for regression discontinuity treatment effect inference where the discontinuity point is unknown. In particularly relevant work, Herlands et al. (2018) define an automated RD search procedure called local regression discontinuity discovery (LoRD3) that first requires fitting a "smooth" background function to the probability of treatment, and then computing test statistics for candidate RDs based on the residuals of the background function. However, neither LoRD3 nor Porter and Yu (2015)'s approach consider heterogeneity in treatment assignment. Our method is the first RD discovery algorithm that frames heterogeneity in additional covariates which affect treatment as a machine learning task.

Other bodies of work consider treatment *effect* discovery (McFowland III et al., 2018; Ogallo et al., 2021) and CATE estimation for posthoc subgroup analysis (Spiess and Syrgkanis, 2021; Lipkovich et al., 2017), which require randomized controlled experiments with interventions. Though there are some parallels with our work, our method focuses on the treatment *assignment* estimation problem (in contrast to the treatment *effect* estimation problem), making our method amenable to

the discovery of causal inference opportunities agnostic of the downstream outcome. Furthermore, our method only requires observational data that are amenable to regression discontinuity analysis.

Within the regression discontinuity literature, recent work examines the use of covariates in RDs, highlighting the potential for efficiency gains in inference (Cattaneo et al., 2022) and tree-based CATE estimation for a given RD cutoff (Reguly, 2021). Our method advances this space by framing RD study *discovery* as a data-driven task that utilizes covariates to uncover the most promising RD candidates.

Specifically, we take a novel approach by: 1) formulating the discovery procedure as *treatment assignment uptake* (TAU) estimation and 2) identifying heterogeneous subgroups with a higher effective sample size for both improved interpretability and statistical power. Our method improves on prior work by using ML to identify statistically powerful interpretable subgroups for RD studies.

## 3  RD Discovery Framework

In the following section we build upon well-established (and Nobel-prize awarded) econometric estimation frameworks (Imbens, 2014; Angrist and Pischke, 2008) as well as the conditional average treatment effect (CATE) estimation literature (Athey and Imbens, 2016; Chernozhukov et al., 2018) to frame our RD discovery procedure. We then target the effective sample size of the discovered RD and show how optimizing this quantity increases statistical power.

### 3.1  Regression Discontinuity Preliminaries

Here we review the potential outcomes framework for analyzing regression discontinuities (RDs); see, e.g. Imbens and Lemieux (2007) and Cattaneo et al. (2019a) for comprehensive overviews of RDs. We define the following notation for an individual $i$:

$$X_i = \text{running variable for individual } i$$
$$c = \text{assignment threshold for } X_i$$
$$\vec{W}_i = \text{vector of pre-treatment covariates for individual } i$$
$$Z_i = \mathbf{1}[X_i \geq c], \text{ threshold indicator for individual } i$$
$$Y_i = \text{observed outcome for individual } i$$
$$T_i(\cdot) = \text{potential treatment assignment for individual } i$$
$$Y_i(\cdot) = \text{potential outcome for individual } i$$

We note here that the potential treatment assignments are defined in terms of the threshold indicator $T_i(Z_i)$. $T_i(1)$ corresponds to the potential treatment assignment for $X_i \geq c$, and $T_i(0)$ to $X_i < c$. We focus on the "fuzzy" regression discontinuity (FRD) case, which assumes that the probability of *treatment assignment uptake* jumps at $c$, but not necessarily from 0 to 1 (Hahn et al., 2001) (without loss of generality, we also assume that the jump in probability at the threshold is positive) (Hahn et al., 2001):

**Assumption 1** (FRD). *The limits* $x^- = \lim_{x \uparrow c} E[T|X = x]$ *and* $x^+ = \lim_{x \downarrow c} E[T|X = x]$ *exist, and* $\tau = x^+ - x^-$ *is non-zero,* $0 < \tau < 1$.

We also define *compliers* as individuals for which $T_i(1) > T_i(0)$, namely that they receive the treatment when above the threshold, and do not receive treatment when below the threshold. Under the assumptions of *continuity*, *monotonicity*, and *threshold excludability* (Lee and Lemieux (2009), Appendix B.1), the treatment effect estimate $\gamma$ can be written as a ratio:

$$\gamma = \frac{\lambda}{\tau} = \frac{\lim_{x \downarrow c} E[Y|X=x] - \lim_{x \uparrow c} E[Y|X=x]}{\lim_{x \downarrow c} E[T|X=x] - \lim_{x \uparrow c} E[T|X=x]} \tag{1}$$

Where $\lambda$ is the jump in outcome $Y$ at the cutoff $c$, and $\tau$ is the jump in treatment assignment uptake (TAU).

In practice, Equation 1 is often estimated by fitting two local linear regressions using the discontinuity indicator $Z$ within a bandwidth $h$ to the left and right of the cutoff: a "first-stage" *treatment* regression for $\tau$ and a "second-stage" *outcome* regression for $\lambda$ (Imbens and Lemieux, 2007), with data-driven methods developed to select the bandwidth $h$ (Imbens and Kalyanaraman, 2009).

As a running example throughout the text, we consider evaluating the effect of breast cancer screening age guidelines, which recommend beginning screening at age 40 for women (Oeffinger et al., 2015). Setting the recommended screening age is an important decision that can impact millions of patients in the U.S. However, evaluating the causal impact of a particular screening age through a randomized experiment would be difficult logistically as it would disrupt standards of clinical practice. Because the screening decision $T$ is made based on the continuous variable age $X$ being above the threshold of $c = 40$ years, such a causal question can be evaluated using a regression discontinuity design, and has been done in prior work (Kadiyala and Strumpf, 2016). Additionally, since compliance with the threshold assignment is imperfect (not everyone at the age of 40 will deterministically be screened for breast cancer), the scenario lends itself naturally to the fuzzy RD framework of $\tau$ estimation we consider here.

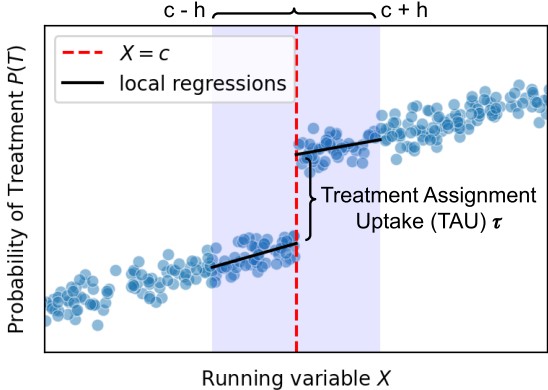

Figure 2: **The anatomy of the RD first stage regression, where a local linear regression is used to estimate TAU $\tau$.** A local linear regression with threshold indicator $Z = \mathbf{1}[X_i \geq c]$ is fit using data within the shaded bandwidth of $X \in [h-c, h+c]$ (Eq. 2). An analogous regression with outcome $Y$ on the vertical axis is used to estimate $\lambda$.

To formulate a data-driven RD discovery procedure, we now turn our attention to the "first-stage" task of estimating $\tau$ in the context of predicting a unit's compliance status.

### 3.2 Treatment Assignment Uptake (TAU)

As the goal of our method is to identify RD opportunities rather than explicitly estimating treatment effects, we focus on estimating treatment assignment uptake (TAU) $\tau$, and in particular will look to maximize TAU using heterogeneity in observed covariates. In our breast cancer example, the TAU increases with greater adherence to the clinical guidelines: the more women who begin screening at the age of 40, the larger the discontinuity is in treatment assignment uptake. $\tau$ is often modeled using a local linear regression within a *bandwidth $h$* around the cutoff $c$ (Hahn et al., 2001):

$$T = \tau_c Z + \beta_0 + \beta_1(1 - Z)(X - c) + \beta_2 Z(X - c) + \epsilon \qquad (2)$$

Where treatment assignment uptake $\tau_c$ is indexed by the cutoff $c$, $\epsilon$ is homoskedastic noise, and samples are within $X_i \in [c - h, c + h]$ (Figure 2). We use this linear probability model estimation strategy in order to ensure causal validity, which is commonly used in the econometric literature as an efficient approach to estimate treatment assignment (though other non-parametric methods can be used as well) (Imbens and Lemieux, 2007). Our RD study discovery task is formalized as a hypothesis test of the existence of a treatment discontinuity at threshold $c$: $H_0 : \tau_c = 0, H_A : \tau_c \neq 0$, which can be tested via significance of the estimated $\hat{\tau}_c$ in Equation 2. Our estimation and subsequent maximization of TAU can be equivalently framed as estimation of compliance probability for subgroups at the threshold (Aronow and Carnegie, 2013) (see Appendix B.2):

**Proposition 1.** *For a given bandwidth $h$ and cutoff $c$, estimating $\hat{\tau}_c$ is equivalent to estimating the probability of compliance $P(T(1) > T(0))$.*

We leverage this connection to maximize TAU heterogeneity and discover the most promising subgroups for RD analysis.

### 3.3 Heterogeneity in TAU

Beyond identifying candidate thresholds $c$ that produce significant TAU estimates, we want to find heterogeneous subgroups among our sample population at a given cutpoint $c$ to propose more statistically powerful RD studies. This problem can be seen as conditional compliance estimation (Kennedy et al., 2020), where we identify the individuals (the compliers) to which the threshold cutoff applies, using the other pre-treatment covariates $\vec{W}$. In our breast cancer screening example, we would clearly want to exclude all men, lest their inclusion reduce the treatment discontinuity due to their non-compliance with the screening guideline. Other factors such as family history or genetic risk may also influence whether individuals adhere to the guideline.

In order to identify such subgroups, we define the heterogeneous TAU estimation task. Given the standard FRD assumptions presented in Section 3.1, Kennedy et al. (2020) and Coussens and Spiess (2021) have shown that estimating the probability a unit is a complier, $\tau_c(\vec{W}_i)$ (their TAU probability), can be framed as conditional average treatment effect (CATE) estimation (Appendix B.3):

**Proposition 2.** *$\tau_c(\vec{W}_i)$ given $c$ can be identified as the conditional probability of compliance.*

$$P(T(1) > T(0)|\vec{W}) = \tau_c(\vec{W}) \qquad (3)$$

Heterogeneous TAU $\tau_c(\vec{W})$ can thus be estimated using data-driven machine learning methods developed in recent years for CATE estimation (e.g., Chernozhukov et al. (2018); Oprescu et al. (2019), and Padilla et al. (2021)). The machine-learned estimates of $\tau_c(\vec{W})$ will be unbiased due to the sample-splitting *honesty* property of such estimators (Athey and Imbens, 2016; Chernozhukov et al., 2018). We can thus estimate $\tau_c(\vec{W})$ for a given RD threshold $c$.

Because our goal is to identify subgroups of individuals where treatment assignment uptake varies, we choose to use tree-based approaches (Oprescu et al., 2019; Athey et al., 2019) for the estimation problem, which provide valid, honest, and interpretable subgroup populations defined by the learned causal tree's nodes. In particular, we can distill a tree-based model that estimates $\hat{\tau}(\vec{W})$ into a

single decision tree, and extract heterogeneous subgroups that correspond to the learned tree's nodes (Athey and Imbens, 2016; Battocchi et al., 2019). Tree-based CATE models thus provide a data-driven approach for identifying interpretable subgroups that have heterogeneous TAU.

### 3.4 From TAU to effective sample size

Though we have established the TAU objective for heterogeneous treatment uptake at a threshold and an approach to identify subgroups using CATE estimators, to actually increase power in finite samples we cannot only account for $\tau_c(\vec{W})$; we also need to consider the sample size of the subgroup. Solely maximizing for TAU when discovering subgroups may not yield higher power, as it is possible for such an objective to select unfeasibly small subgroups with higher TAU: in our breast cancer example, a subgroup of ten women may have a higher TAU than a subgroup of size 1,000 with 50% women, but we would much prefer the latter subgroup in terms of study feasibility. Thus, we propose to target the *effective sample size* (Liu et al., 2022; Heng et al., 2020) of a given subgroup $G$ $\eta_G$, which explicitly accounts for both the TAU as well as the size of the subgroup. Let $P = (\vec{W}_i, Z_i, T_i, X_i)_{i=1}^{N_P}$ represent the "baseline population" samples i.e., all of the samples within the bandwidth $h$ for a cutoff $c$ where $N_P$ is the sample size, and $G = (\vec{W}_j, Z_j, T_j, X_j)_{i=1}^{N_G}$ represent the samples that are part of a subgroup $G$, where $N_G$ is the sample size. The TAU for a subgroup $G$ is defined as $\tau_G := \tau_c(\vec{W}_G)$, where $\vec{W}_G$ are the pre-treatment covariates that define a sample's membership in group $G$; similarly, $\tau_P$ is the TAU for all of the samples in the baseline population. Letting $\mathbb{E}_G[\cdot]$ be expectations over subgroup $G$, the effective sample size $\eta_G$ is then:

$$\eta_G = N_G\tau_G^2 = N_G(\mathbb{E}_G[T|Z=1] - \mathbb{E}_G[T|Z=0])^2 \tag{4}$$

The intuition behind $\eta_G$ is that only units compliant with the threshold indicator $Z$ contribute to the treatment effect estimation. Non-compliers contribute noise to the estimate, so the "effective" sample is the nominal sample size scaled by a quantity of $\tau_G$, which is the probability of compliance with the threshold indicator (Proposition 1). We want to maximize this as the variance of a fuzzy RD estimator will decrease as the effective sample size increases (Coussens and Spiess, 2021; Liu et al., 2022; Heng et al., 2020). Power thus increases as effective sample size increases (Appendix B.4):

**Proposition 3.** *Power is a non-decreasing function of $\eta_G$, regardless of subgroup size $G$.*

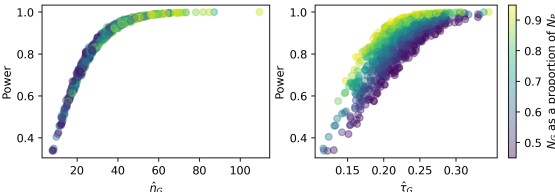

Figure 3: **Optimizing for effective sample size (left) increases statistical power regardless of subgroup sample size.** We simulate 1,000 random subgroups and show power against the effective sample size $\eta_G$ (left) and treatment assignment uptake ($\tau_G$) (right), where the shading indicates different subgroup sizes as a proportion of the total population.

Maximizing effective sample size $\eta$ is a superior objective to maximizing heterogeneous TAU alone as it is possible to select a small subgroup $G$ that has a high TAU but will still have lower power than the baseline population sample. We demonstrate this empirically in Figure 3, which together with Proposition 3 motivates the use of $\eta_G$ in our algorithm.

### 3.5 A test statistic for effective sample size

When discovering subgroups with higher effective sample size than the baseline population, we want to ensure that the differences are not

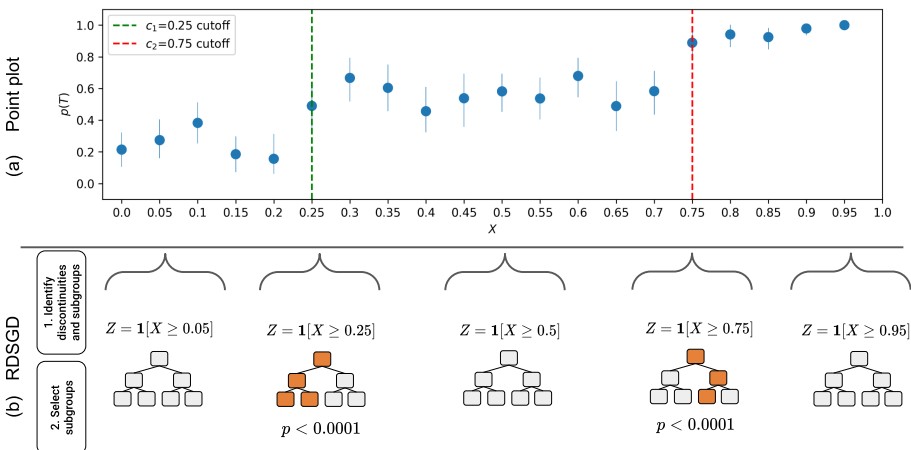

Figure 4: **RDs with heterogeneous cutoffs have smaller TAU (a), but RDSGD can correctly identify cutoffs with hetereogeneity (b).** a) treatment probabilities across running variables $X$ (95% CIs). b) representation of RDSGD (Algorithm 1), where causal trees are fit to each candidate threshold $Z$ which generate subgroups with higher effective sample sizes (step 1) and statistically significant subgroups are selected (orange nodes, step 2).

due to noise in the selected samples. We thus test if the effective sample size for a subgroup $G$ is greater than that of the whole population $P$: $H_0 : \eta_G - \eta_P = 0, H_A : \eta_G - \eta_P > 0$. The corresponding test statistic is:

$$t_\eta = \frac{\eta_G - \eta_P}{\sqrt{\text{Var}[\eta_G - \eta_P]}} \tag{5}$$

Though $\text{Var}[\eta_G - \eta_P]$ can be difficult to derive as groups $G$ and $P$ are overlapping, we leverage properties of influence functions (Newey and McFadden, 1994; Kahn, 2022) to construct a consistent estimator for this variance term (Appendix B.5). As empirical $\eta$ (Equation 4) can be easily calculated using sample means, we can estimate $t_\eta$. We verify this test statistic behaves correctly asymptotically under the null hypothesis (Figure B.1). We can thus leverage heterogeneity in treatment assignment uptake for improved study power.

## 4 Methodology

We calculate the effective sample size presented in Section 3 to implement our **RDSGD** (Regression Discontinuity SubGroup Discovery) method, which is outlined in Algorithm 1 and visualized in Figure 4. RDSGD comprises of two main steps: 1) identification of candidate thresholds and subgroups with higher effective sample size, and 2) subsequent selection of subgroups[1]. As we use a data-driven process to select the subgroups to test (Kuchibhotla et al., 2022; Athey and Imbens, 2016), to enable valid inference in this two-step procedure we assume the existence of two separate datasets $S_1, S_2$ through sample splitting or a holdout set.

---

[1]Source code and the data needed to reproduce all figures are available at: `https://github.com/tliu526/rdsgd`.

### 4.1 Identifying Discontinuities and Subgroups

To discover regression discontinuities with potential heterogeneity, we first identify candidate thresholds. Given a set of cutpoints $C_X = \{c_1, c_2, ...\}$ for a running variable $X$, RDSGD analyzes thresholds $c \in C_X$. It first generates threshold indicator $Z := \mathbf{1}[X \geq c]$ and selects a bandwidth $h_c$ (Algorithm 1, step 1a), which can be chosen by the user or by a data-driven selection process (Cattaneo et al., 2019a; Imbens and Kalyanaraman, 2009).

---

Algorithm 1: RD SubGroup Discovery (RDSGD)

1. **Identify discontinuities and subgroups.**

   For $c \in C_X$:
   (a) Select bandwidth $h_c$ of analysis and generate threshold indicator $Z := \mathbf{1}[X \geq c]$
   (b) Select baseline population $P = \{(\vec{W}_i, Z_i, T_i, X_i) \mid i \in [c - h_c, c + h_c]\}$ from $S_1$ and compute effective sample size $\hat{\eta}_P$
   (c) Fit subgroup tree model $\hat{f}$ estimating $\hat{\tau}(\vec{W})$ (Eq. 3)
   (d) Obtain subgroups $G_{s,c} = \{(\vec{W}_i, Z_i, T_i, X_i) \mid i \in s\}$ from $S_1$ and $\hat{\eta}_{G_s}$ for each node $s$ in $\hat{f}$
   (e) Output subgroups with stat. sig. greater effective sample size $G_c = \{G_{s,c} \mid \hat{\eta}_{G_s} > \hat{\eta}_P\}$

2. **Select subgroups.**

   (a) For each subgroup definition $G_{s,c} \in \bigcup_{c \in C_X} G_c$:
       i. Select data $X_G = \{X_j \mid (j \in G_{s,c})\}$ from $S_2$
       ii. Fit local linear estimator $\hat{T}(X_G, c)$ (Eq. 2), obtain TAU estimate $\hat{\tau}_G$ and p-value $p_{\hat{\tau}_G}$
   (b) Compute corrected significance level $\tilde{\alpha}$
   (c) Output discovered cutoffs and subgroups: $D_X = \{(c, G_{s,c}) \mid (p_{\hat{\tau}_G} < \tilde{\alpha})\}$

---

RDSGD then computes the baseline effective sample size $\hat{\eta}_P$ (step 1b) from data sample $S_1$. Because $X$ is real-valued, it can theoretically yield infinite potential cutpoints. However, in many situations (such as the clinical contexts we consider) the grid of candidate cutpoints for $X$ can be sensibly defined in terms of the running variable e.g., in whole years for age-based clinical guidelines or at the precision of lab result readings. Where the candidate cutpoints do not have a sensible definition, other methods such as LoRD3 (Herlands et al., 2018) can be used to provide $C_X$ (Section 2).

Next, RDSGD generates subgroups for each $c$ based on the pre-treatment covariates $\vec{W}$ by estimating $\hat{\tau}_c(\vec{W})$ from $S_1$ for the given cutoff and bandwidth (Steps 1c-d). As discussed in Section 3.3, RDSGD uses a causal tree approach to estimate $\hat{\tau}_c(\vec{W})$ and produces candidate subgroups $G_{s,c}$ for a given cutoff $c$ for RD study evaluation. RDSGD then determines if the subgroup effective sample size $\hat{\eta}_{G,s}$ (Section 3.5) is statistically greater than the baseline $\hat{\eta}_P$ (Step 1e).

### 4.2 Selecting subgroups

Once we have candidate heterogeneous subgroups, we need to select the most promising subgroups in terms of study power while preserving statistical validity by using the separate sample $S_2$ to test. Given a subgroup $G_{s,c}$ for a cutpoint $c$, RDSGD evaluates the local linear regression for TAU to test for the discontinuous jump indicative of a potential RD study (Step 2a). In order for the TAU test to be valid, RDSGD must account for the multiple comparisons across the set $G_X$ of *all*

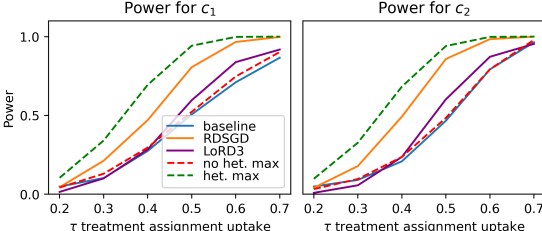

(a) **RDSGD improves the statistical power of discovering RD opportunities by considering heterogeneity.** We simulate RDs over 500 trials for each $\tau$ and record the number of correct discoveries at $c_1$ and $c_2$ for an empirical power estimate.

(b) **RDSGD discovers more powerful subgroups over baseline in higher dimensions.** We run 500 trials with $\tau = 0.5$, recording mean power and comparing with baseline powers.

| $\dim(\vec{W})$ | $c_1$ power | $c_2$ power |
|---|---|---|
| baseline | 0.52 | 0.48 |
| 2 | **0.79**$\pm$0.15 | **0.78**$\pm$0.15 |
| 4 | **0.78**$\pm$0.15 | **0.77**$\pm$0.16 |
| 8 | **0.79**$\pm$0.16 | **0.78**$\pm$0.16 |
| 16 | **0.78**$\pm$0.15 | **0.78**$\pm$0.15 |

Figure 5: **Synthetic experiments demonstrate the benefit of RDSGD.**

the subgroups considered for running variable $X$, including the candidate subgroups generated in Step 1d. RDSGD thus applies a Bonferroni correction to produce the adjusted significance level $\tilde{\alpha}$ (Step 2b). Finally, RDSGD outputs discovered subgroups and cutoffs based on $\tilde{\alpha}$ (Step 2c). By leveraging connections between 1) TAU estimation and machine-learned CATE estimation as well as 2) our statistical testing framework for effective sample size (Sections 3.4-3.5), RDSGD is a data-driven RD discovery procedure that uses potential heterogeneity among subgroups to identify more powerful RD studies (Algorithm 1, Figure 4).

## 5 Synthetic Experiments

We first validate RDSGD using synthetic data where multiple discontinuities in a given running variable can be distinguished via heterogeneity in other observable covariates. We compare RDSGD to a baseline method that only tests the TAU regression of Equation 2 for each cutpoint $c$ (Algorithm A) and thus does not consider heterogeneity. We also make comparisons to the LoRD3 method proposed by Herlands et al. (2018). Full simulation details can be found in Appendix C.

### 5.1 Heterogeneity in One Covariate

**Data Generation.** Here we generate data where half of the units in our sample follow a fuzzy RD threshold for running variable $X \in [0, 1]$ at $c_1 = 0.25$, $Z = \mathbf{1}[X \geq c_1]$, while the other half follow a fuzzy RD threshold at $c_2 = 0.75$, $Z = \mathbf{1}[X \geq c_2]$. The threshold a particular unit follows can be identified by observed covariate $W \in [0, 1]$, with units $W < 0.50$ following threshold $c_1$ and units $W \geq 0.50$ following threshold $c_2$ (Appendix C.1-C.3). Such a scenario might arise in real-world settings where the clinical threshold varies depending on other patient attributes; from our running breast cancer example, women with high risk of breast cancer due to hereditary factors ($W = 1$) should begin screening earlier than the recommended age of 40 for women without risk factors ($W = 0$) (Center for Disease Control, 2021). The TAUs at $c_1$ and $c_2$ will appear much smaller if covariate $W$ is not accounted for, thus this synthetic data scenario is one where we would expect RDSGD to be useful.

**Power Calculations.** To quantify RDSGD's performance, we calculate the theoretical power that can be achieved for a given RD study. Given the regression framework for TAU estimation (Equation 2), we analytically derive the theoretically achievable power levels (Appendix B.6). We can then use these power calculations and our synthetic data to evaluate the baseline method, LoRD3, and RDSGD. We simulate RD datasets as described above, evaluating empirical power for each ground-truth $\tau \in [0.2, 0.3, ..., 0.7]$, to correctly identify discontinuities at $c_1$ and $c_2$.

**Simulated results.** Our empirical results show the benefit of RDSGD (Figure 5a). We also calculate the theoretical power without considering heterogeneity in $W$ (red dashed lines) and find that the baseline method (Algorithm A.1, blue lines) matches that power level across $\tau$. RDSGD (Algorithm 1, orange lines) improves upon the baseline method as well as LoRD3 (Appendix C.4). RDSGD maintains empirical false positive rates below the nominal $\alpha = 0.05$ for all $\tau$ levels due to the multiple testing corrections (Figure C.3). Empirical power for RDSGD approaches the theoretical power when heterogeneity in $W$ is accounted for (dashed green lines). The gap between the power levels of RDSGD and the theoretical power is expected, as we lose power due to testing corrections and the data-driven tree fitting, which does not perform an exhaustive search.

## 5.2 Heterogeneity in Multiple Covariates

The improvement in power over baseline also extends to multidimensional heterogeneity (Table 5b), where we increase the dimensionality of the covariates $\dim(\vec{W}) \in [2, 4, 8, 16]$ that determine whether an individual complies with cutoff $c_1$ or $c_2$ and record the power of the discovered subgroups (Appendix C.5). Though power (as expected) decreases slightly as $\dim(\vec{W})$ increases, RDSGD scales well to higher dimensions, as the average subgroup powers for both cutoffs across all $\dim(\vec{W})$ are greater than the baseline theoretical powers. These simulation results (Figure 5a, Table 5b) provide empirical evidence that RDSGD can improve RD discovery in the presence of heterogeneity.

## 6 Case Study: Medical Claims Data

To evaluate RDSGD in real-world settings, we target a variety of clinical contexts where we believe RDs exist: breast cancer screening, colon cancer screening, and diabetes diagnosis. We use Optum's de-identified Clinformatics® Data Mart Database (2007-2018), which contains claims data on diagnoses, procedures, prescriptions, and lab results. We use all adults with demographic data, roughly 60 million unique patients over the twelve-year period. We note that each clinical setting uses a different subset as there are inclusion criteria that are specific to that setting, e.g. the presence of a lab result. Full details on demographics and sample selection can be found in Appendix D.

### 6.1 Data Extraction and Featurization

For each clinical setting, we target a specific running variable $X$ that corresponds to a treatment $T$ (see the first two columns of Table 1). We use LOINC, CPT, and ICD codes (McDonald et al., 2003; WHO, 2016) to identify lab results, procedures, and diagnoses. To convert longitudinal data in the claims database into tabular form, we index a patient by the first recorded presence of the running variable (Appendix D.2). We then assume a fixed time window after the running variable is recorded for the treatment to occur, in order to account for lags in claims data reporting. If the treatment of interest appears for the patient within the window, they are coded as "treated," otherwise they

Table 1: **Discovered RD thresholds and subgroups in medical claims data.** We report $\hat{\tau}$ and $\hat{\eta}$ (higher is better in both cases) for baseline RDs and discovered subgroups.

| Clinical guideline | Running variable | Threshold | Subgroup discovered | Baseline $\hat{\tau}$ (SE) | RDSGD $\hat{\tau}$ (SE) | Baseline $\hat{\eta}$ | RDSGD $\hat{\eta}$ |
|---|---|---|---|---|---|---|---|
| Breast cancer screening | Age | $\geq 40$ | Gender = Female | 0.039 (0.0013) | **0.088 (0.003)** | 471.7 | **1074.6** |
| Colon cancer screening | Age | $\geq 50$ | Encounter date > 2014-05-07 | 0.062 (0.002) | **0.12 (0.002)** | 1335.1 | **1759.0** |
| Type 2 diabetes diagnosis | A1C % | $\geq 6.5$ | Encounter date > 2010-06-07 | 0.093 (0.005) | **0.12 (0.006)** | 1212.1 | **1533.6** |

are "untreated." For example, in the setting where we wish to estimate the "treatment" uptake of diabetes diagnosis, we find a patient's first recorded A1C measurement and then search the database for a Type II diabetes diagnosis within the following week. In the screening settings where age is the running variable, we use a patient's age at their first preventative care visit.

We additionally query the claims database for covariates that may impact TAU heterogeneity. For all clinical settings, we consider heterogeneity across patient demographics (age, gender, race), socio-economic status (education level, household income) and claims-specific features (initial encounter date, insurance type). Our data extraction method provides a pipeline for converting raw claims data into feature matrices for RDSGD.

We use the EconML package (Battocchi et al., 2019) to estimate the heterogeneous TAU tree model needed for RDSGD (Algorithm 1, step 1d). We split our data into equally sized samples $S_1, S_2$ for each clinical context. Due to our larger sample size, most candidate RDs that RDSGD returns will have power approaching 1, so here we compare the estimated TAU and effective sample size $\eta$ of the discovered subgroups to the baseline RD without considering heterogeneity.

## 6.2 Results

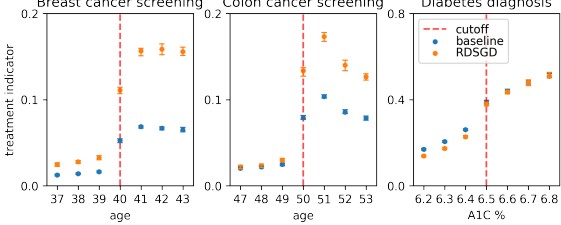

Figure 6: **RDSGD discovers subgroups that improve TAU in different clinical settings.** We show the discovered cutoff and the probability of treatment for the entire sample (blue) and for subgroup discovered by RDSGD (orange).

The most promising discovered RD thresholds and subgroups for each clinical setting are shown in Table 1, with treatment probability point plots shown in Figure 6.

**Breast Cancer.** RDSGD correctly identifies that breast cancer screening only applies to women age 40 (Oeffinger et al., 2015), doubling the effective sample size from 471.7 to 1074.6.

**Colon Cancer.** RDSGD correctly identifies the recommended screening age of 50 for colon cancer, and additionally discovered a subgroup of patients who were more likely to be screened at the threshold; these individuals had an encounter date later than 2014-05-07, producing a subgroup with a higher effective sample size than

the baseline population (1335.1 vs. 1759.0). This could be due to increased adherence to screening from a guideline update that occurred approximately in the same time period (US Preventive Services Task Force, 2016).

**Type 2 Diabetes.** RDSGD identifies the A1C cutoff of 6.5% for diabetes diagnosis (American Diabetes Association, 2010), and also identifies a subgroup more likely to be compliant with the cutoff, increasing the effective sample size from 1212.1 to 1533.6. This subgroup excludes patients who have encounters before 2010-06-07, which aligns with clinical practice intuition as A1C was not introduced as a diagnostic criteria until 2010.

## 7 Discussion

Here we have proposed RDSGD, a method for regression discontinuity (RD) discovery that produces interpretable subgroups by optimizing for the *effective sample size* through a machine learning framework. We demonstrate through synthetic studies how RDSGD provides power improvements in the presence of heterogeneity. We apply RDSGD to a variety of clinical settings, both validating our method as well as discovering new RDs to investigate. We now discuss how our method fits into real-world workflows as well as highlight limitations and future work.

### 7.1 Real-World Workflows

RDSGD is most useful in scenarios for treatment effect estimation when explicit randomization of the treatment is not possible. While our method only discovers RD opportunities within the data (the so-called treatment assignment regression, Section 3.1) and does not make treatment *effect* estimates, the primary goal is to identify quasi-experimental randomness that can be used to estimate the downstream effects of the given treatment $T$ on an outcome of interest $Y$. For example, in our A1C diabetes case study, a practitioner may wish to study the effect of the A1C cutoff on different outcomes, such as metformin prescription rate or heart attack incidence. By identifying both the cutoff as well as subgroups where the effective sample size is stronger, any downstream treatment effect estimation a practitioner wishes to conduct has both: 1) an identified variable that provides quasi-experimental randomness and 2) an interpretable cohort to which it applies. By performing causal inference opportunity *discovery*, our method increases research efficiency and scalability by identifying the most promising RD studies to pursue.

We encourage researchers to think about RDSGD as a scientific discovery tool: instead of driving a manual process where practitioners have to generate RD candidates based on domain expertise alone, RDSGD produces candidates from the data which the practitioners can then verify. This not only accelerates research using larger observational data as it is much easier to verify potential candidates rather than generate them, but also provides a means to increase research efficiency. For a researcher, there is opportunity cost in terms of time and effort when analyzing an RD, and our method can inform which studies they should pursue, as well as which studies they potentially should not pursue. For example, Naidech et al. (2020) investigates a seemingly promising RD opportunity in stroke guidelines, but due to sample size and compliance issues had inconclusive results. RDSGD could have complementary utility by providing an early signal to practitioners on which opportunities may not ultimately be fruitful if their hypothesized cutoff does not appear as a subgroup. We note that the "bottom-up" data-driven approach we advocate for may reduce the power of subsequent studies that use the discovered design if the design was known apriori due to

the use of sample splitting. However, we believe this tradeoff is worthwhile in many situations when RD designs are unknown to identify the most promising opportunities for new causal studies.

We believe that our method is well-positioned to discover RDs that depend on a moderate number of covariates – this is particularly relevant within the medical domain as clinical practice evolves to become more nuanced. To give an example in existing work, Scott et al. 2021 [6] use cardiovascular risk score cutoffs as an RD design to study the effect of statins on adverse outcomes; they perform manual analogues of our subgroup method by 1) identifying hospitals that have better adherence to the score cutoff guidelines and 2) by excluding patients with very high cardiovascular risk and comorbidities that might interfere with the treatment discontinuity e.g., patients with diabetes are not recommended to be prescribed statins. Furthermore, as we show in our case studies (Section 6), when working with longitudinal datasets it is possible that the candidate RD cutoffs may shift over time due to changes in clinical best practice, and it is important to surface such shifts to the practitioner. We note that these subgroup examples we give here are relatively low dimensional. We want to caution that highly complex RDs that are found via our data-driven discovery approach might have limited empirical value, and warrant careful attention before being categorized as valid studies. Nevertheless, RD subgroups that are defined by a moderate number of covariates may still be missed if only the running variable is considered in isolation from the other pre-treatment variables, and it is these scenarios that RDSGD fits well.

Furthermore, RDSGD could be used to investigate implicit differences in treatment assignment. Because the subgroups produced by RDSGD define clear inclusion criteria based on the path of the fitted causal tree, it can be used to identify sources of bias in treatment decisions such as those documented in FitzGerald and Hurst (2017); Hausmann et al. (2013); Hoffman et al. (2016).

## 7.2 Limitations and Future Work

We highlight some limitations and opportunities for future work. First, we do not make treatment effect estimates as part of our method and defer that step to practitioners, who are free to choose which outcome $Y$ they wish to study. Care must be taken when moving forward to treatment effect estimation ($\gamma$, Section 3.1). When making treatment effect estimates with identified cutoffs we need to be mindful of the *exclusion restriction* assumption Imbens and Lemieux (2007), which can be violated when the cutoff decision $Z$ affects the outcome $Y$ outside of its affects on $T$. Critically, validation tests specific to RDs, such as whether the running variable has been manipulated and continuity in the covariates, also need to be run to ensure an appropriate causal design (McCrary, 2008; Imbens and Lemieux, 2007). It is straightforward to apply tests of RD validity post our method, and they are important verification steps a practitioner needs to conduct after selecting a discovered RD subgroup to evaluate.

Moreover, there are limitations in using medical claims data, as there may be selection biases in healthcare utilization as well as potential under-reporting of diagnoses and treatments of interest (Jensen et al., 2015). Thus, when moving forward with RD studies identified by RDSGD, practitioners need to work closely with domain experts to ensure that causal validity is maintained.

We also note that because RDSGD uses causal tree-based methods to identify heterogeneous TAU subgroups, it is inherently greedy (see Figure 5a where RDSGD approaches, but does not achieve max power). Though our use of trees was a deliberate design decision made to maintain interpretability and scalabililty to large datasets (Wu et al., 2022), future work could investigate other

methods that are optimal in terms of TAU maximization: for example, applying policy learning methods that maximize power in randomized trials to RDs (Spiess and Syrgkanis, 2021). Furthermore, as we use sample splitting to ensure valid inference, additional efficiency gains could be achieved in extensions through cross-fitting, conditional inference, or other bias correction methods (Kuchibhotla et al., 2022; Zhao et al., 2023).

Though we have focused on clinical use cases in our motivation and case study, we emphasize that RDSGD can be utilized in any settings where treatment assignment heterogeneity in RD thresholds may exist, such as in education (Chay et al., 2005; Goodman, 2008), political science (Klašnja and Titiunik, 2017), and labor market programs (Lalive, 2007), among others (Cattaneo et al., 2016). Future work could look to apply RDSGD to these contexts. Our method could also be extended to identify regression *kink* designs, another related framework popular in economics, where there is a kinkpoint rather than a discontinuity at treatment threshold (Card et al., 2015).

### 7.3  Conclusion

Here we introduce RDSGD, a machine-learning method for RD discovery which identifies interpretable subgroups with higher effective sample size, increasing study feasibility. RDSGD is effective on both simulated and real data, and could provide new avenues for more credible observational causal studies across medicine and social science through quasi-experimental designs.

### Statement of Broader Impact

Data-driven RD discovery promises to enable more studies that estimate treatment effects from observational data, providing valuable causal estimates when conducting a randomized experiment would be expensive, unethical, or infeasible. However, there is some risk in using our method, shared with much of ML, as it can leverage and propagate biases in the data. For example, if a demographic indicator predicts non-compliance, whether true or due to e.g. dataset bias, the treatment effect estimates made in RD studies utilizing our method would exclude the non-compliant group. This could disadvantage that group as the study would not apply to them. The discovered subgroups need to be scrutinized for potentially perpetuating bias, and one advantage of our method is the transparency of to whom the discovered thresholds apply to. Healthcare data are of particular interest, which we utilize as examples throughout this work, and disparities against minority groups in such data are common (Murthy et al., 2004). Data could also include selection biases (such as demographic variation in healthcare utilization resulting in differences in collected claims data), and we must be mindful of these sources of bias when interpreting results. To mitigate these potential harms, practitioners should carefully consider the source of their data and whenever possible verify their results with a different dataset e.g., claims data from a different healthcare system. When using data-driven methods like RDSGD for causal analysis from observational data, stakeholders must weigh the strength of evidence supporting the causal claim as well as the result's external validity before making decisions based off of the study.

### Acknowledgments

We thank Morgan McGuire for valuable discussion and feedback. We thank the anonymous reviewers whose detailed feedback have improved this work.

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

## A  Baseline algorithm details

We give the full baseline RD discovery procedure used for comparison with RDSGD in Algorithm A.1.

---

Algorithm A.1: Baseline method: RD threshold discovery

---

1. For $c \in C_X$:

    (a) Select bandwidth $h_c$ for treatment regression (Section 3.1)
    (b) Select data $X = \{X_i \mid X_i \in [c - h_c, c + h_c]\}$
    (c) Fit estimator $\hat{T}(X, c)$ of Equation 2 to obtain TAU estimate $\hat{\tau}_c$ and output p-value $p_{\hat{\tau}_c}$

2. Compute corrected corrected significance level $\tilde{\alpha} = \alpha/|C_X|$

3. Output discovered cutoffs and bandwidths: $D_X = \{(c, h_c) \mid p_{\hat{\tau}_c} < \tilde{\alpha}\}$

## B  Mathematical details

### B.1  FRD and IV assumptions

We assume the following standard FRD assumptions for valid estimation (Lee and Lemieux, 2009) to identify Equation 1:

- **Continuity**. Both the potential outcomes $Y(1)$ and $Y(0)$ are continuous as a function of the running variable $X$:

$$E[Y_i(1)|X], E[Y_i(0)|X] \text{ continuous over domain of } X \tag{6}$$

- **Monotonicity**. $X$ crossing the cutoff cannot simultaneously cause some units to take up and others to reject the treatment (also known as the "no defiers" assumption):

$$T(1) \geq T(0) \tag{7}$$

- **Excludability of crossing the threshold.** $X$ crossing the cutoff cannot impact $Y$ except through impacting the treatment:

$$Y(T = t, Z = z) = Y(T = t) \tag{8}$$

    Where $Y(T = t, Z = z)$ is the potential outcome to be observed if $T = t$ and $Z = z$.

Given the equivalence between FRDs and IVs (Imbens, 2014), two-stage least squares (TSLS) estimation uses analogous assumptions:

- **Consistency**. If $Z = z$ and $T = t$, then the observed outcomes of $T$ and $Y$ are the potential outcomes under $Z = z$ and $T = t$.

$$T = ZT(1) + (1 - Z)T(0) \tag{9}$$
$$Y = TY(1) + (1 - T)Y(0) \tag{10}$$

- **Unconfounded instrument**. The instrument is unconfounded with the potential treatment given the observed covariates.

$$Z \perp T(1), T(0) \mid \vec{W} \tag{11}$$

- **Monotonicity**. The instrument cannot simultaneously cause some units to take up and others to reject the treatment.

$$T(1) \geq T(0) \tag{12}$$

- **Exclusion restriction**. The instrument cannot impact $Y$ except through the treatment:

$$Y(T = t, Z = z) = Y(T = t) \tag{13}$$

### B.2   Proposition 1 details: compliance and TAU equivalence

Imbens and Lemieux (2007); Hahn et al. (2001) show how a fuzzy regression discontinuity estimated via local linear regression for a given cutoff $c$ and fixed bandwidth $h$ is numerically equivalent to a two-stage least squares (TSLS) estimation problem with the following additional regressors:

$$\begin{pmatrix} 1 \\ \mathbf{1}[X_i < c](X_i - c) \\ \mathbf{1}[X_i \geq c](X_i - c) \end{pmatrix} \tag{14}$$

Note that the instrument $Z_i$ is defined as $Z_i = \mathbf{1}[X_i \geq c]$, the same as our RD cutoff indicator. These regressors thus give the form of the treatment regression in Equation 2.

Imbens and Rubin (2015) further show that under IV assumptions of consistency, unconfounded instrument and monotonicity (Appendix B.1), the treatment effect estimate can be expressed as:

$$\gamma_{IV} = \frac{E[Y_i(1) - Y_i(0)|\text{unit i is a complier}] \cdot \pi_{\text{comply}}}{\pi_{\text{comply}}} \tag{15}$$

Where $P(T(1) > T(0)) = \pi_{\text{comply}}$ is the probability of compliance. Within the data bandwidth of analysis $h$, we can equivalently write $\tau$ of Equation 1 in terms of $Z$ (Hahn et al., 2001):

$$
\begin{aligned}
\tau &= E[T|Z = 1] - E[T|Z = 0] \\
&= P(T = 1|Z = 1) - P(T = 1|Z = 0) \\
&= 1 - P(T = 0|Z = 1) - P(T = 1|Z = 0) \\
&= 1 - \pi_{\text{never taker}} - \pi_{\text{always taker}}, \text{ (unconfounded instrument)} \\
&= \pi_{\text{comply}}, \text{ (monotonicity)}
\end{aligned} \tag{16}
$$

Where $\pi_{\text{always taker}}$ and $\pi_{\text{never taker}}$ are the proportions of always-takers and never-takers. We then have that $\tau$ is equivalent to the probability of compliance, and that the first stage (denominator of $\gamma_{IV}$ from Equation 15) regression of the TSLS framework estimates the probability of compliance. Thus we can use the TSLS framework for our analysis and estimation of RD TAU.

### B.3 Proposition 2 details: conditional compliance identification

Identification of the conditional probability of compliance follows a similar argument as Appendix B.2. Given the cutoff choice $c$ generating $Z = \mathbf{1}[X \geq c]$ and a fixed bandwidth $h$, we can use the equivalent analysis of $Z$ as an IV like we do in Appendix B.2. The conditional probability of compliance is given by:

$$P\left(T(1) > T(0)|\vec{W}\right)$$

From monotonicity, there are no defiers (units where $T(1) < T(0)$). We can thus write:

$$
\begin{aligned}
&P(T(1) > T(0)|\vec{W}) \\
&= 1 - P\left(T(1) = 1, T(0) = 1|\vec{W}\right) - P\left(T(1) = 0, T(0) = 0|\vec{W}\right)
\end{aligned}
\tag{17}
$$

The latter two terms are the probability of always-takers and never-takers given covariates $\vec{W}$, respectively. From the unconfounded instrument assumption, these quantities can be *identified* (converted from causal quantities to estimable statistical quantities) (Imbens and Rubin, 2015):

$$P(T(1) = 1, T(0) = 1|\vec{W}) = P(T = 1|Z = 0, \vec{W})$$
$$P(T(1) = 0, T(0) = 0|\vec{W}) = P(T = 0|Z = 1, \vec{W})$$

This then gives:

$$
\begin{aligned}
P(T(1) > T(0)|\vec{W}) &= 1 - P(T = 1|Z = 0, \vec{W}) - P(T = 0|Z = 1, \vec{W}) \\
&= 1 - P(T = 1|Z = 0, \vec{W}) - (1 - P(T = 1|Z = 1, \vec{W})) \\
&= P(T = 1|Z = 1, \vec{W}) - P(T = 1|Z = 0, \vec{W})
\end{aligned}
\tag{18}
$$

Allowing us to identify the conditional probability of compliance, and thus the conditional TAU $\tau_c(\vec{W})$ as desired.

**Equivalence of CATE and heterogeneity TAU.** Next, we make a connection between the conditional TAU and conditional average treatment estimation (CATE).

The CATE of a treatment $T$ on outcome $Y$ is given by:

$$\text{CATE} = E[Y|T = 1, \vec{W}] - E[Y|T = 0, \vec{W}] \tag{19}$$

Given the two-stage process of RD treatment effect estimation, we not only have potential outcomes $Y(\cdot)$ but also potential treatments $T(\cdot)$. We can thus equivalently analyze the "treatment effect" the cutoff indicator $Z$ has on "outcome" $T$, yielding Equation 3. The same standard fuzzy RD

assumptions that enable estimation of the treatment effect $\gamma$ at cutoff $c$ (Section 3.1) also enable estimation of the heterogeneous TAU $\tau_c(\vec{W})$ through CATE estimation frameworks (Kennedy et al., 2020; Coussens and Spiess, 2021).

### B.4 Proposition 3 details: power and effective sample size

We describe the relationship between $\eta$ and power. From Cattaneo et al. (2019b), we have the following power function for an $\alpha$-level two-sided test for $\tau$ given a local linear treatment regression:

$$\beta(\tau) = 1 + \Phi\left(\frac{\tau}{\sqrt{V}} - z_{\alpha/2}\right) - \Phi\left(\frac{\tau}{\sqrt{V}} + z_{\alpha/2}\right) \tag{20}$$

where $\Phi$ is the Normal distribution CDF, $z_t$ is its $t$th percentile (e.g. $z_a = \Phi^{-1}(a)$), and $V$ is the variance of $\tau$. Next, leveraging the equivalence between instrumental variable (IV) analysis and fuzzy regression discontintuities (FRD) established in Proposition 1, the variance of an FRD estimator under constant treatment effects has been shown to be (Coussens and Spiess, 2021):

$$V = \frac{\mathrm{Var}[Y|Z, \text{ compliers}]}{\eta E[Z](1 - E[Z])} \tag{21}$$

Thus, variance decreases as $\eta$ increases. We note that even under the relaxation of the constant treatment effects assumption, Heng et al. (2020) and Baiocchi et al. (2014) have shown that the IV variance with $n$ samples is at least as large as the variance with $\eta$ samples of known compliers. Thus, the two-sided power function is non-decreasing as $V$ decreases, and hence when $\eta$ increases. Freeman et al. (2013) also equivalently show this relationship between IV power and $\eta$ in their power calculation analysis of Mendelian randomization studies.

Note that our statements are of the power non-decreasing as a function of $\tau$ because power is bounded ($\beta(\tau) \in [0, 1]$). We show empirically in Figure 3 that power in practice increases as $\hat{\eta}$ increases, regardless of the size of the subgroup. Data simulated in Figure B.1 follow the TAU regression data generation described in Appendix C.1 with $n = 1000$, $\tau = 0.2$, and a bandwidth of 0.5. Random subgroups of sizes uniformly distributed between 450 and 950 are drawn to show the relationship between $\tau_G$ and $\eta_G$ and power across varying $G$ sizes.

### B.5 Effective sample size test statistic derivation

From Section 3.5, in order to construct a test statistic for Equation 5 we need a consistent estimator of $\mathrm{Var}[\eta_G - \eta_P]$:

$$\mathrm{Var}[\eta_G - \eta_P] = \mathrm{Var}[\eta_G] + \mathrm{Var}[\eta_P] - 2\mathrm{Cov}[\eta_G, \eta_P] \tag{22}$$

We use *influence functions* to empirically estimate this variance term (Newey and McFadden, 1994). To derive the influence function for $\eta_G$, we first give the influence function of a sample $i$ on $\tau_G$, which can be seen as a difference-in-means estimator (Imbens and Rubin, 2015; Kahn, 2022):

$$\psi_{\tau_G,i} = \mathbf{1}[Z_i = 1]\frac{N_G}{N_{1G}}(T_i - \overline{T}_{1G}) - \mathbf{1}[Z_i = 0]\frac{N_G}{N_{0G}}(T_i - \overline{T}_{0G}) \tag{23}$$

Where $\mathbf{1}[\cdot]$ is the indicator function and $N_{zG}$ is the number of samples in subgroup $G$ where $Z = z$. We can then apply the influence function chain rule (Kahn, 2022) to obtain the influence function for $\eta_G$. For an estimator $\hat{\theta}$ such that $\hat{\theta} = T(\hat{\theta}_j, ..., \hat{\theta}_n)$, the influence function of $\hat{\theta}$ is:

$$\psi_{\hat{\theta},i} = \sum_j^n \frac{\partial T}{\partial \hat{\theta}_i} \psi_{\hat{\theta}_j,i} \tag{24}$$

The influence function $\psi_{\eta_G,i}$ of $\eta_G$ for a given sample in subgroup $G$ is thus:

$$\begin{aligned}
\psi_{\eta_G,i} &= \frac{\partial}{\partial \hat{\tau}_G} N_G \hat{\tau}_G^2 \\
&= 2 N_G \hat{\tau}_G \psi_{\tau_G,i} \\
&= 2 N_G \hat{\tau}_G \left( \mathbf{1}[Z_i = 1] \frac{N_G}{N_{1G}} (T_i - \overline{T}_{1G}) - \mathbf{1}[Z_i = 0] \frac{N_G}{N_{0G}} (T_i - \overline{T}_{0G}) \right)
\end{aligned} \tag{25}$$

Note that in following text we refer to $\psi_{\eta_G,i}$ as $\psi_{G,i}$ to reduce notational clutter.

$\psi_{G,i}$ consists of products and differences of empirical means over different subgroups which are straightforward and fast to compute. Following the properties of influence functions (Newey and McFadden, 1994; Erickson and Whited, 2002), we can next derive the variance-covariance matrix of $\eta_P, \eta_G$ as follows. Let:

$$\Psi = \left[ \vec{\psi}_P, \vec{\psi}_G \right]_{N_P \times 2} \tag{26}$$

where $\vec{\psi}_G$ is a vector of length $N_P$ with values at the $i$th index of $\psi_{G,i}$ if $i \in G$ and 0 otherwise. As the distribution of an estimator $\theta$ is equivalent to $\frac{1}{\sqrt{N}} \sum_i^N \psi_{\theta,i}$ (Erickson and Whited, 2002), we have:

$$\sqrt{N_P} \begin{pmatrix} \hat{\eta}_P - \eta_P \\ \hat{\eta}_G - \eta_G \end{pmatrix} = \frac{1}{\sqrt{N_P}} \sum_i^{N_P} \begin{pmatrix} \psi_{\eta_P,i} \\ \psi_{\eta_G,i} \end{pmatrix} \xrightarrow{d} \mathcal{N} \left( \begin{pmatrix} 0 \\ 0 \end{pmatrix}, \mathbb{E} \begin{pmatrix} \psi_{\eta_P,i}^2 & \psi_{\eta_P,i}\psi_{\eta_G,i} \\ \psi_{\eta_P,i}\psi_{\eta_G,i} & \psi_{\eta_G,i}^2 \end{pmatrix} \right) \tag{27}$$

Kahn (2022) then shows that the following variance-covariance matrix gives us the variances of $\eta_G$, $\eta_P$ on the diagonal and the covariance of $\eta_G$ and $\eta_P$ on the off-diagonal:

$$V = \frac{1}{N_P^2} \left( \Psi^T \Psi \right) \tag{28}$$

The elements of $V$ give us empirical, consistent estimates of $\mathrm{Var}[\eta_G]$, $\mathrm{Var}[\eta_P]$, and $\mathrm{Cov}[\eta_G, \eta_P]$, allowing us to calculate $t_\eta$ (Equation 5).

As a sanity check, we verify that the distribution of the computed test statistic under the null hypothesis behaves correctly in empirical simulations (Figure B.1). Data simulated in Figure B.1 follow the TAU regression data generation described in Appendix C.1 with $n = 200$, $\tau = 0.5$, and a bandwidth of 0.5. Random overlapping subgroups of size 100 are drawn to test the null distribution.

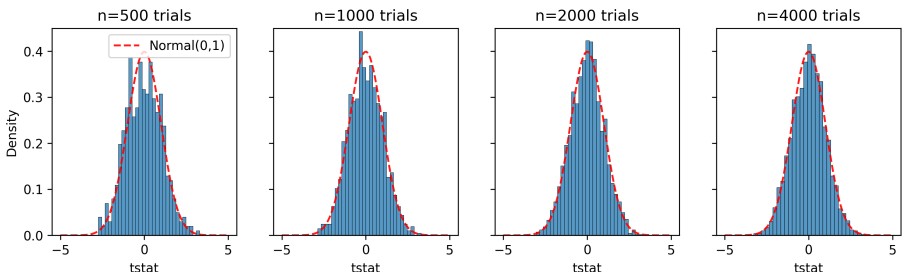

Figure B.1: Our overlapping $\eta$ hypothesis test produces well-behaved null distributions, yielding valid p-values. We conduct six different simulations with varying numbers of trials under the null hypothesis that the overlapping groups have the same effective sample size.

### B.6 Closed-form power calculations

We use Equation 20 shown in Appendix B.6 to compute the theoretical power of a treatment regression. To calculate $V$, Imbens and Lemieux (2007) give a closed form solution for the asymptotic variance of the treatment regression, assuming a symmetric bandwidth $h$:

$$V = \frac{8 \cdot p_{\text{bw}}}{n}(\sigma^2_{T,l} + \sigma^2_{T,u}) \tag{29}$$

where $n$ is the total sample size, $\sigma^2_{T,l}$ is the TAU variance below the cutoff, $\sigma^2_{T,u}$ is the TAU variance above the cutoff, and $p_{\text{bw}}$ is the fraction of units in the sample in the bandwidth $h$.

Since $T$ is binary, we have that:

$$\sigma^2_{T,l} = \lim_{x\uparrow c} Var(T|X = x) = \mu_{T,l} \cdot (1 - \mu_{T,l})$$
$$\sigma^2_{T,u} = \lim_{x\downarrow c} Var(T|X = x) = \mu_{T,u} \cdot (1 - \mu_{T,u})$$

where $\mu_{T,l} = \lim_{x\uparrow c} Pr(T = 1|X = x)$ and $\mu_{T,u} = \lim_{x\downarrow c} Pr(T = 1|X = x)$. $\mu_{T,u}$ and $\mu_{T,l}$ can be computed given our simulated data generating process, giving us a closed form solution of the theoretical power that can be achieved in our synthetic experiments.

## C Synthetic experiment details

### C.1 TAU regression setup

For our blended RD simulation scenario, data are generated in the following manner. Let $X_i \sim \text{Unif}(0,1)$ be the running variable for unit $i$. We generate threshold indicator $Z_i = \mathbf{1}[X_i > c]$, where $c$ is the chosen treatment threshold. Each unit's probability of treatment assignment is defined as:

$$p_i = \tau Z_i + \nu X_i + \eta + \psi_i \tag{30}$$

where $\tau$ is the true TAU, $\nu$ the coefficient determining the running variable's effect on the outcome, $\eta$ a constant, and $\psi_i$ a Gaussian noise term. For each generated data set, we vary $\tau$ and draw

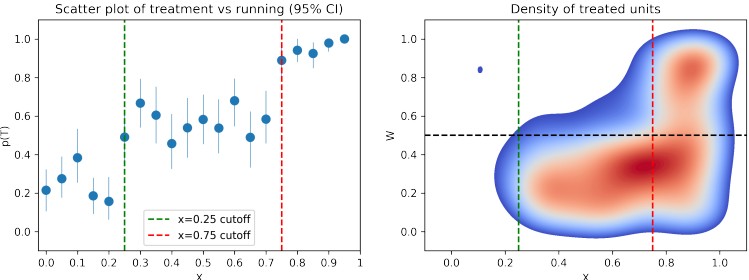

Figure C.2: **Heterogeneity in TAU can be observed when considering the additional covariate $W$.** Left is a pointplot of TAU probabilities as a function of the running variable $X$ (the same as pictured in Figure 4) for a single trial of our simulation, with error bars as 95% CIs and a nominal $\tau = 0.7$. Right is a density plot of treated units as a function of both $X$ and $W$.

$\nu \sim N(0, 0.1)$ and set $\eta = 0.2, \nu = 0.05$. $p_i$ values are clamped to $[0, 1]$, with an individual's treatment assignment then defined as $T_i \sim \text{Bern}(p_i)$.

## C.2  Blended RD in one covariate

For the simulation setting shown in Section 5, for each unit we additionally draw $W_i \sim \text{Unif}(0, 1)$, with the cutoff $c$ a unit complies with being determined by:

$$c = \begin{cases} 0.25 \text{ if } W_i < 0.5 \\ 0.75 \text{ if } W_i \geq 0.5 \end{cases} \tag{31}$$

For each trial of our synthetic experiment, we draw 1000 units for each sample $S_1, S_2$ with half of the units complying with the lower cutoff and half complying with the upper cutoff. Differences in TAU can be observed when visualizing heterogeneity in $W_i$ (Figure C.2).

## C.3  Treatment assignment uptake reduction in the presence of heterogeneity

As our simulated data contains two equally-sized populations that comply with different cutoffs, the observed $\tau$ will effectively be reduced by half, since for each cutoff $c_1, c_2$, half of the units do not comply with the jump in TAU at that point. For example, at a given $\tau$ and $n$, we can estimate the observed TAU at $c_1$ for our sample by computing $\mu_{T,u} - \mu_{T,l}$:

$$\mu_{T,u} = \lim_{x \downarrow c_1} Pr(T = 1 | X = x)$$

$$= \frac{\nu \cdot c_1 \cdot n + \eta \cdot n + \left(\tau \cdot \frac{n}{2}\right)}{n} = \nu c_1 + \eta + \frac{1}{2}\tau$$

$$\mu_{T,l} = \lim_{x \uparrow c_1} Pr(T = 1 | X = x)$$

$$= \frac{\nu \cdot c_1 \cdot n + \eta \cdot n}{n} = \nu c_1 + \eta$$

Thus we have $\mu_{T,u} - \mu_{T,l} = \frac{1}{2}\tau$.

## C.4 Heterogeneity in one covariate: simulation details

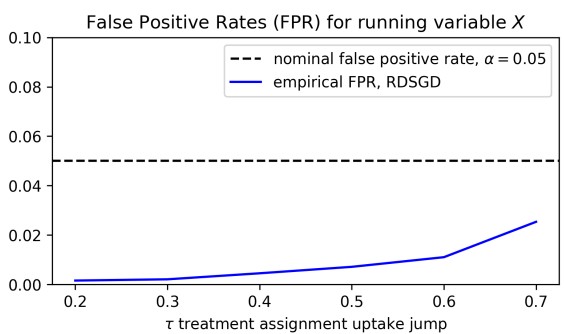

Figure C.3: **RDSGD maintains false positive rates below the nominal $\alpha$ level.** We simulate RDs with multiple running variables over 500 trials for each $\tau$ and record the number of false discoveries.

Results presented in Section 5.1 use data generated according to Appendix C.1-C.2. We use a fixed bandwidth $h = 0.25$ to ensure that the closed form theoretical powers calculated in Appendix B.6 are valid. Causal forests were fit according to default parameters specified in the EconML package (Battocchi et al., 2019) (with *honesty* enabled for valid and unbiased inference), and a fixed depth of 3 and minimum leaf size of 100 were used for subsequent CATE causal trees distilled from the forests to ensure subgroups remained interpretable. We note that the causal forest implementation in EconML by default runs a two-fold cross validation internally when selecting hyperparameters for the LogisticRegressionCV scikit-learn models for treatment, which searches over L2 regularization parameters in a grid of 10 values between $1e-4$ and $1e4$ using the default accuracy criterion. Seeds were passed to machine learning models to ensure reproducibility. All 500 trials were seeded with their trial number, and once implementation was complete experiments were run twice to validate reproducibility. We run Herlands et al. (2018)'s RD discovery method, LoRD3, according to recommended parameters in their code repository, setting (in their notation) $k = 100$ and $z = \{X, W\}$ so that information from both $X$ and $W$ are used. All simulations were run on a Ubuntu 20.04 LTS server, with a 24-core Intel i9-7920X CPU and 94 GB RAM.

The false positive rates shown in Figure C.3 are computed based on the number of significant discontinuities discovered that do not equal $c_1$ or $c_2$ divided by the total number of tests over the 500 trials for each $\tau$ level. As our empirical power metric amounts to a count of "successful" detections of statistically significant discontinuities at $c_1$ and $c_2$ out of the 500 trials, we use a $\chi^2$ test (or corresponding Fisher's exact test if count numbers are not sufficient) to compare RDSGD's performance with the baseline discovery method. All the differences between RDSGD and the baseline algorithm are statistically significant with $p < 0.001$ With the exception of the $\tau = 0.2$ case, and the differences between LoRD3 and RDSGD are statistically significant at all $\tau$ levels.

## C.5 Heterogeneity in multiple covariates: simulation details

Results presented in Section 5.2 used data generated according to Appendix C.1. Instead of having a single covariate govern the choice of $c$ as in Appendix C.2, we generate $\dim(\vec{W})$ covariates $W_1, W_2, ...$ for each unit $i$ using scikit-learn's make_regression() method, producing output $\omega_i$. We then scale $\omega_i$ to fall in the range $[0, 1]$ with sample mean 0.5, and determine which cutoff a unit complies with by:

$$c = \begin{cases} 0.25 \text{ if } \omega_i < 0.5 \\ 0.75 \text{ if } \omega_i \geq 0.5 \end{cases} \tag{32}$$

We fix ground truth $\tau = 0.5$ for the simulation trials and calculate the baseline oracle power according to Appendix B.6. All hyperparameters, hardware, and seeding strategy for the 500 trials are the same as described in Appendix C.4.

All statistically significant subgroups discovered by RDSGD are recorded, and their power is computed according to Equation 20. Means and standard deviations reported in Table 5b are taken across all trials. For significance testing, we use a one-sample t-test comparing the subgroups in each cell of Table 5b with their corresponding baseline oracle powers. All tests were statistically significant at $p < 0.001$.

## D    Clinical setting and cohort details

### D.1    Justifying use of private claims dataset

In order to evaluate RDSGD in a real-world setting where our clinical collaborators can help verify discovered candidate RD studies, we needed a large-scale clinical data source that spanned general healthcare settings with enough data granularity on individuals so our method can leverage potential TAU heterogeneity across common demographic covariates (described in Appendix D.2). The claims dataset that we use has the advantage of having an array of disease classes and visibility into patient information in general healthcare settings, as opposed to publicly available datasets such as MIMIC which focuses on a very specific context (critical care). Working with such detailed patient information necessitates adherence to federal HIPAA privacy rules concerning privacy, which restrict access to "protectable health information"; we do however provide descriptive statistics of the presented in Table D.1 as well as anonymized datasets that are sufficient to recreate all figures in this paper.

### D.2    Data extraction per clinical setting

**Breast cancer screening**. We extract a patient's first recorded routine preventative care visit as designated by ICD and CPT codes. The treatment indicator $T$ for a patient is whether they received a breast cancer screen as designated by ICD code within 7 days of the recorded encounter date. The running variable $X$ is the patient's age at the initial encounter date (note that in order to protect patient privacy, the claims database only has resolution to a patient's year of birth). We consider candidate thresholds of $C_X = [40, 45, ..., 60]$ with the data-driven bandwidth selected to be 4, at age increments of 5 years to align with typical screening guideline values.

**Colon cancer screening**. Similar to the breast cancer setting, we use a patient's first recorded routine preventative care visit. The treatment indicator $T$ for a patient is whether they received a colon cancer screen within 7 days of the recorded encounter date, and age is the running variable $X$ with candidate thresholds of $C_X = [40, 45, ..., 60]$ and bandwidth 4, at age increments of 5 years to align with typical screening guideline values.

**Type 2 diabetes diagnosis**. We extract patient's first recorded A1C measure as designated by LOINC codes and use it as the running variable $X$. The treatment indicator $T$ for a patient

is whether a type II diabetes diagnosis ICD code appears in their record within 30 days of the first recorded A1C measure. We consider candidate thresholds of $C_X = [5.0, 5.1, 5.2, ..., 7.5]$ and a data-driven bandwidth selected to be 0.4 as this is the standard range of A1C values, with the lab readings having precision to one decimal place.

In all three clinical settings, we exclude patients that have a recorded treatment indicator code prior to their initial encounter date, as well as patients that do not have recorded demographic information. The following covariates are included as $\vec{W}$ for each patient (unordered categorical variables are one-hot encoded, while ordinal variables are coded as integers): gender, encounter date, insurance type (Medicare vs commercial), race, education level, and household income range.

### D.3 Full result and cohort details

Claims data analyses were run on a secure CentOS Linux 7 server with a 40-core Intel Xeon E5-4650 CPU and 504 GB RAM. We follow the same hyperparameter and model training strategy as described in Appendix C.4. For the RD candidate cutoff identified in each clinical setting, we show the best subgroup in terms of effective sample size in Table 1. We describe the demographics of patients within the RD bandwidth of analysis in Table D.1.

Table D.1: **Demographic details for each clinical setting within RD bandwidth.**

|  | Breast cancer screen, age $\geq 40$ | Colon cancer screen, age $\geq 50$ | Type 2 diabetes diagnosis, A1C $\geq 6.5$ |
|---|---|---|---|
| Sample size | 315,225 | 346,191 | 133,826 |
| Mean age (SD) | 39.9 (1.98) | 49.9 (1.98) | 59.6 (13.4) |
| Gender (%) |  |  |  |
|    Male | 169,050 (53.6) | 171,601 (49.6) | 61,583 (46.0) |
|    Female | 146,175 (46.4) | 174,590 (50.4) | 72,242 (54.0) |
| Race (%) |  |  |  |
|    White | 212,441 (67.4) | 247,169 (74.3) | 77,945 (58.2) |
|    Black | 32,700 (10.4) | 33,387 (9.6) | 23,159 (17.3) |
|    Asian | 26,158 (8.3) | 16,578 (4.8) | 10,481 (7.8) |
|    Hispanic | 43,926 (13.9) | 39,057 (11.3) | 22,241 (16.6) |
| Education level (%) |  |  |  |
|    Less than 12th grade | 1,285 (0.4) | 1,349 (0.4) | 1,098 (0.8) |
|    High school diploma | 59,943 (19.0) | 73,995 (21.4) | 40,783 (30.5) |
|    Less than Bachelor's | 166,612 (52.9) | 187,171 (54.1) | 71,527 (53.4) |
|    Bachelor's degree plus | 87,385 (27.7) | 83,676 (24.2) | 20,418 (15.3) |
| Household income (%) |  |  |  |
|    <$40k | 47,529 (15.1) | 50,651 (14.6) | 35,456 (26.5) |
|    $40k - 49k | 17,756 (5.6) | 19,290 (5.6) | 11,312 (8.5) |
|    $50k - 59k | 18,388 (5.8) | 21,196 (6.1) | 12,234 (9.1) |
|    $60k - 74k | 28,648 (9.1) | 33,265 (9.6) | 15,938 (12.0) |
|    $75k - 99k | 47,319 (15.0) | 56,234 (16.2) | 21,417 (16.0) |
|    >$100k | 155,585 (49.4) | 165,555 (47.8) | 37,469 (27.9) |

