# OpenReview forum: "Automated Detection of Causal Inference Opportunities: Regression Discontinuity Subgroup Discovery"
_TMLR — Accepted by TMLR_

### Review · Reviewer_14oM · 2023-07-12

**Summary Of Contributions:**

  In this paper, the authors develop Regression Discontinuity Subgroup Discovery (RDSGD), which is a machine learning method that identifies RD thresholds for different subgroups.

The main insight of the paper is that for a particular conjectured threshold in a regression discontinuity design, searching for subgroups with a large change in treatment uptake probabilities can be recast as a compliance prediction problem (i.e., a conditional average treatment effect on the potential treatment function Di()). The authors therefore can apply ideas from heterogeneous treatment effect estimation to search for heterogeneity across subgroups in treatment response probabilities at the conjectured threshold using ideas from Kennedy et al. (2020) and Coussens & Spiess (2021).

Practically, for a considered running variable and discrete grid, the authors in Algorithm 1 consider a search procedure for combinations of thresholds and subgroups that are have large treatment takeup effects. Then, given the groups and thresholds with the largest such effects, the authors estimate the report a corrected significance level (Bonferroni correction) of the change in treatment uptake and return the thresholds and subgroups with the most significant effects. The authors apply their method to a large real-world dataset on medical claims, and show that RDSGD returns known thresholds that meaningfully affect treatment decisions and disease risk.

**Audience:**

Yes

**Broader Impact Concerns:**

I see no broader impact concerns.

**Claims And Evidence:**

Yes

**Requested Changes:**

See the discussed weakness above.

**Strengths And Weaknesses:**

Strength #1: The RGSD procedure is simple to implement and leverages off-the-shelf tools for summarizing treatment effect heterogeneity to discover combinations of thresholds and subgroups. As a result, there are no new technical results in the paper, but that is ok. The paper's contribution is to combine existing results on heterogeneous treatment effect estimation to the problem of causal discovery for RD designs.

Strength #2: The authors illustrate their procedure in a real medical claims dataset and extensive simulations. The methods appear to be practically useful given these experiments.

Weakness #1: In selecting subgroups (step #2), RDSGD should construct the test statistics and p-values on a hold out sample. Otherwise, the selected thresholds + groups will suffer from an post-selection inference problem (e.g., Berk et al. 2013 and many others). In other words, because the data are used twice -- first to select which hypothesis to test and second test the selected hypotheses -- the resulting tests will not control size correctly.

Alternatively, the authors could try to characterize the selection event and try to apply recent results on conditional inference (e.g., Lee et al. 2016) but that seems difficult since the first step of identifying thresholds and subgroups is quite complex. Sample splitting seems like a simple solution. The conditional inference problem here does share some similarities to the problem of doing inference after estimating a break point in econometrics (e.g., Andrews et al. 2021).

Question #1: Could these ideas be simply extended to regression kink designs? E.g., similar to Porter & Yu (2015) that consider estimating a regression discontinuity with an unknown threshold, Hansen (2017) does the same for regression kink designs (i.e., breaks in the first-derivative of the conditional expectation function).

References:
Berk et al. 2013: Valid post-selection inference
Lee et al. 2016: Exact post-selection inference with application to Lasso.
Andrews et al. 2021: Inference After Estimation of Breaks
Hansen (2017): Regression Kink With an Unknown Threshold

---

> ### Author Response · Authors · 2023-09-01
> **Response to Reviewer 14oM**
>
> We appreciate your positive and constructive feedback, and respond in order below:
>
> **Weakness 1**
>
> Thank you for highlighting this important point, it is much appreciated. We make the modifications necessary to our algorithm for sample splitting and re-generate our results – we find that the conclusions from our experiments remain the same, with our method performing well in identifying subgroups with higher effective sample size over baseline methods in both simulated and real data settings. Critically, the discovered subgroups for our three clinical case studies remained quite consistent with our prior results. We also add to Section 7.2 a discussion of future work that could extend our method with cross-fitting, conditional inference, or other bias correction techniques [1, 2].
>
> **Question 1**
>
> We agree that our method could be extended to kink designs, as there are fuzzy analogues of kink designs where (non)-compliance with the kink may vary for some samples [3]. We have updated the discussion in 7.2 to highlight this avenue for future work.
>
> **References**
>
> [1] Kuchibhotla et al. 2022. Post-Selection Inference. Annual Review of Statistics and Its Application.
>
> [2] Zhao et al. 2023. Inference on subgroups identified based on a heterogeneous treatment effect in a post hoc analysis of a clinical trial. Clinical Trials.
>
> [3] Card et al. 2015. Inference on Causal Effects in a Generalized Regression Kink Design. Econometrica.

---

### Review · Reviewer_ifu5 · 2023-08-16

**Summary Of Contributions:**

**Summary.** The authors propose a method (RDSGD) which identifies subgroups in a set of covariates $X$. For each subgroup, the authors aim to maximise a regression discontinuity (RD), which manifests whenever a strict threshold determines treatment assignment. RDSGD is interesting for situations where the threshold is not known prior to inferring a treatment effect in RDs + for when it is reasonable to assume there can be different thresholds in different areas in $X\in\mathcal{X}$.



**Audience:**

Yes

**Broader Impact Concerns:**

n.a.

**Claims And Evidence:**

Yes

**Requested Changes:**

My requested changes are to clarify my weaknesses/questions above, in the main text or the appendix. I am also open to not include these clarifications, but I would need strong motivation from the authors for not doing so.

**Strengths And Weaknesses:**

### Strengths

* I like the different perspective of this work within treatment effects. While I am quite unfamiliar with RDs, this seems like a novel persective, at least in machine learning, which mostly focusses on inference, rather than clustering (?).
* Overall the paper looks good. There are some illustrative figures, examples and intuitive experiments. There are a few points which I think could still be improved (listed below).
* I look forward to follow-up work.

### Weaknesses

* Being unfamiliar with RDs, I think a more gentle introduction would help at least me in reading this paper. Introducing RD in 3.1 is done rather compact. Since TMLR is a journal (rather than a conference), I would encourage the authors to spend a bit more space on this. If the main text is too limited, I would definitely advise in favour of a detailed appendix.
* I would really enjoy a running example. The authors mention diabetes a few times, but I did not find that very helpful. Would it be possible to provide an example where standard treatment effect inference methods are insufficient and one better uses an RD method? Furthermore, would it be possible to use the example to differentiate RDSGDs objective (of finding subgroups) is superior to standard RD methods?
* cfr. my question below, I think it would also be beneficial to discuss (albeit briefly) the (C)ATE field in general. Currently, RDs are presented as an island of literature but I would prefer to see it in relation to more well-known methods in treatment effects.


### Questions
* I am quite unfamiliar with IDs for treatment effects. Perhaps you can clarify, for me, how having a _deterministic_ assignment policy ($Z_i = \mathbf{1}[X_i > c_i]$, aligns with typical TE assumptions such as overlap (which assumes _anyone_ can receive treatment, including $X_i < c_i$).

---

> ### Author Response · Authors · 2023-09-01
> **Response to Reviewer ifu5**
>
> Thank you for the positive and constructive comments. We respond to your feedback in order below:
>
> **Weakness 1**
>
> We have added more details introducing RDs in Section 3.1, specifically including more context on the fuzzy RD assumption and how treatment effect estimation is performed in practice. We also include an additional Figure 2 visualizing the first stage estimation process.
>
> **Weakness 2**
>
> We utilized a breast cancer screening example for motivating the search for subgroups maximizing effective sample size (Section 3.3) as well as the synthetic experiment setup (Section 5.1) in our initial submission. We have updated the text in the following places to carry this as a running example throughout:
>
> - Section 3.1: As a running example throughout the rest of the text, we consider evaluating the effect of breast cancer screening age guidelines, which recommend beginning screening at age 40 for women [1]. Setting the recommended screening age is an important decision that can impact millions of patients in the U.S. However, evaluating the causal impact of a particular screening age through a randomized experiment would be difficult logistically as it would disrupt standards of clinical practice. Because the screening decision $T$ is made based on the continuous variable age $X$ being above the threshold of $c=40$ years, such a causal question can be evaluated using a regression discontinuity design, and has been done in prior work [2]. Additionally, since compliance with the threshold assignment is imperfect (not everyone at the age of 40 will deterministically be screened for breast cancer), the scenario lends itself naturally to the fuzzy RD framework of $\tau$ estimation we consider here.
>
> - Section 3.2: In our breast cancer example, the TAU increases with greater adherence to the clinical guidelines: the more women who begin screening at the age of 40, the larger the discontinuity is in treatment assignment uptake.
>
> - Section 3.3: In our breast cancer screening example, we would clearly want to exclude all men, lest their inclusion reduce the treatment discontinuity due to their non-compliance with the screening guideline. Other factors such as family history or genetic risk may also influence whether individuals adhere to the guideline.
>
> - Section 3.4: …Solely maximizing for TAU when discovering subgroups may not yield higher power, as it is possible for such an objective to select unfeasibly small subgroups with higher TAU: in our breast cancer example, a subgroup of ten women may have a higher TAU than a subgroup of size 1,000 with 50\% women, but we would much prefer the latter subgroup in terms of study feasibility.
>
> - Section 5.1: from our running example, women with high risk of breast cancer due to hereditary factors ($W=1$) should begin screening earlier than the recommended age of 40 for women without risk factors ($W=0$) [3]. The TAUs at $c_1$ and $c_2$ will appear much smaller if covariate $W$ is not accounted for…
>
>
> **Weakness 3**
>
> We have added additional discussion of related work in Section 2, in particular highlighting the CATE subgroup literature as well as the use of covariates in RDs and positioning our proposed method within these areas.
>
> **Question 1**
>
> You are correct in this observation; the threshold assignment indicator violates the assumption of overlap (specifically in sharp RDs). Thus, instead of overlap RD designs require an assumption of continuity around the threshold (Appendix B.1, Equation 6), where the treatment effect at the threshold can be estimated via units with running variable values arbitrarily close to above and below the cutoff [4, 5]. The other assumptions needed for identification in the fuzzy RD case are closely related to the identification assumptions of instrumental variables, which we discuss in Appendix B.1-3.
>
> **References**
>
> [1] Oeffinger et al. 2015. Breast Cancer Screening for Women at Average Risk: 2015 Guideline Update From the American Cancer Society. JAMA.
>
> [2] Kadiyala and Strumpf 2016. How effective is population-based cancer screening? Regression discontinuity estimates from the US guideline screening initiation ages. Forum for Health Economics and Policy.
>
> [3] Center for Disease Control 2021. Strategies for Managing Risk: Preventive Screening Recommendations for Women at Increased Risk for Hereditary Breast and Ovarian Cancer.
>
> [4] Lee and Lemieux 2010. Regression Discontinuity Designs in Economics. Journal of Economic Literature.
>
> [5] Imbens and Lemieux 2008. Regression Discontinuity Designs: A Guide to Practice. Journal of Econometrics.

---

### Review · Reviewer_Kxby · 2023-08-23

**Summary Of Contributions:**

The authors provide a novel approach for discovering regression discontinuity designs in observational data. Specifically, their approach is distinguished by the ability to detect high-powered, coherent subsets of data for an RD. The paper provides both synthetic and real world experiments validating the approach and showing the potential for improving the speed and nuance of scientific and social scientific research.

**Audience:**

Yes

**Broader Impact Concerns:**

No concerns here.

**Claims And Evidence:**

Yes

**Requested Changes:**

My requested changes follow the weaknesses I noted above:
1) Not critical but highly recommended.
2) Critical.
3) Critical.
4) Not critical but would love the author's thoughts.
5) Not critical.
6) Potentially critical.

**Strengths And Weaknesses:**

1) Very clear exposition with great appendix materials
2) Novel approach even conceptually to conduct a subgroup search and optimize /tau. Very clever and intuitively appealing
3) Important work on the effective subgroup size, a nice approach that can inspire future work and allow for methodological comparison
4) Good combination of synthetic and real world results with strong comparisons to relevant baseline methods. Great to see the approach discovering real world RDs in a complex dataset.

Weakness:
1) The authors utilize a hypothesis testing framework (and correction for multiple hypothesis testing). However, there is no discussion about what it means to have an experimental design with 95% confidence. How does this impact subsequent results utilizing that experimental design? In traditional econometric research, the RD is not discovered programmatically but by reason/intuition. While this might not be scalable, there is no computational concern that the RD reduces the power of the subsequent research. How would a researcher think about utilizing the results from this model for their research?

2) In their real world experiments the authors note relatively "simple" subgroups with  only 1 dimension. That is reasonable since much more complicated subgroups would quickly become uninterpretable and thus not very useful. From a different perspective, you can say that even if the algorithm discovered an RD in high dimensions, we would be skeptical it is a true RD, since RDs are traditionally thought of as artifacts of human/bureaucratic rules. These are, by definition simple or dumb rules, not likely to be discovered in high dimensional searches.
Said simply: what is the econometric value of an RD discovered across multiple covariates? Are there are practical examples/papers you can point to that utilize such an RD?

3) In section 7.2 the authors briefly mention that they have not considered testing "whether the running variable has been manipulated" as per McCrary, 2008. However, I believe LoRD3 runs that test. Did the authors ensure to modify LoRD3 to not run that test as to ensure equitable comparison of the results between the methods? Or did they take another approach?

4) In section 7.2 the authors briefly mention that they have not considered testing "whether the running variable has been manipulated" as per McCrary, 2008. I have a concern that perhaps the subgroup optimization, especially around effective sample size, can somehow cause the algorithm to falsely identify an RD due to bunching, or else produce a subgroup that falsely passes a bunching test. Honestly I can't think of a good example here but it was concerning that this wasn't addressed in the paper as bunching is a quite common issue with RDs in practice.

5) It seems that LoRD3 performs less testing than RDSGD. Thus in a "simple" fuzzy RD we would expect LoRD3 to have more power than RDSGD. Is that correct? The synthetic experiments presented in section 5 assume a subgroup-based RD. Thus it's reasonable that RDSGD outperforms all baselines. However, what would be the results for synthetic experiments with a simple fuzzy RD? It would be useful to understand not only where RDSDG outperforms, but also where it underperforms (if ever).

6) A bit disappointing to only have 1 real world case study, especially since the results were not very surprising/insightful from a practitioner point of view.

(I really like the article and the author's approach. If any of my critiques are too harsh, it is from a place of excitement about the paper and I hope the authors will forgive any slight)

---

> ### Author Response · Authors · 2023-09-01
> **Response to Reviewer Kxby part 1**
>
> Thank you for your detailed and constructive comments, no slight here! We respond to the numbered questions below in the following comments:
>
> **Question 1**
>
> The goal of our method is the *discovery* of well-powered first stage designs, but it is not necessarily a testing scenario. Our main purpose of using a multiple testing framework in the first stage design is to minimize false discoveries of subgroups. These first stage RD opportunities can then be used to evaluate the treatment effects on any downstream outcome, where inference and testing is more critical – higher power in the first stage is mainly an indicator that any downstream study is more likely to succeed, which is important in practice as RDs can be inconclusive due to sample size or power concerns; we discuss this in paragraphs 3-4 of Section 1 in our submission.
>
> To your second point on researchers utilizing results from this model, researchers should think about our method as a scientific discovery tool: instead of driving a manual process where practitioners have to generate RD candidates based on domain expertise alone, RDSGD produces candidates from the data which the practitioners can then verify. This not only accelerates research using larger observational data as it is much easier to verify potential candidates rather than generate them, but also provides a means to increase research efficiency. For a researcher, there is opportunity cost in terms of time and effort when analyzing an RD, and our method can inform which studies they should pursue, as well as which studies they potentially should not pursue. For example, Naidech et al. 2020 [1] investigates a seemingly promising RD opportunity in stroke guidelines, but due to sample size and compliance issues had inconclusive results. RDSGD could have complementary utility by providing an early signal to practitioners on which opportunities may not ultimately be fruitful if their hypothesized cutoff does not appear as a subgroup. Your questions bring up good points about the framing of our method, and so we have added additional discussion on this topic in Section 7.1.
>
> **References**
>
> [1] Naidech et al. 2020. Probing the Effective Treatment Thresholds for Alteplase in Acute Ischemic Stroke With Regression Discontinuity Designs. Frontiers in Neurology.

---

> ### Author Response · Authors · 2023-09-01
> **Response to Reviewer Kxby part 2**
>
> **Question 2**
>
> We believe that our method is well-positioned to discover RDs that depend on multiple covariates - this is particularly relevant within the medical domain as clinical practice evolves to become more nuanced. For example, there is emerging evidence that demographic factors should be considered for differential guidelines for type 2 diabetes screening. The BMI cutoff for diabetes screening in the general population is 25 kg/m^2, but further research [1] has additionally led to the recommendation that Asian Americans should be screened at 23 kg/m^2 [2]. There have been advocacy campaigns to raise awareness for this particular recommendation [3] so it is certainly possible that even this relatively simple rule may not be known by a practitioner. There have also been resolutions passed at the U.S. state level (Hawaii, California, Massachusetts, among others) [4] that would further introduce geographic location as a factor to condition the potential RD on.
>
> To give other examples in existing work, Kane 2003 [5] evaluates the impact of financial aid grants on college enrollment, where the aid decision depends on discontinuities in multiple factors: GPA, total assets, and household income. Scott et al. 2021 [6] use cardiovascular risk score cutoffs to study the effect of statins on adverse outcomes; they perform manual analogues of our subgroup method by 1) identifying hospitals that have better adherence to the score cutoff guidelines and 2) by excluding patients with very high cardiovascular risk and comorbidities that might interfere with the treatment discontinuity e.g., patients with diabetes are not recommended to be prescribed statins.
>
> Furthermore, as we show in our case studies, when working with longitudinal datasets it is possible that the candidate RD cutoffs may shift over time due to changes in clinical best practice, and it is important to surface such shifts to the practitioner. For example, see our discussion of the discovered subgroups for Type 2 diabetes diagnosis and colon cancer screening in Section 6.2 where encounter dates describe a subgroup that aligns with changes in clinical guidelines. Though we cannot analyze the diabetes screening guideline in our dataset as BMI does not appear in medical claims data, since the recommendation was made in 2015 [2] one could expect that an RD candidate would be present at the cutoff of 23 conditioning on race = Asian AND encounter date > 2015 AND state resolution = True.
>
> To your point about interpretability, we use a tree-based approach precisely to address the potential issues of uninterpretable subgroups. These subgroup examples we give here are still relatively low dimensional, and we agree with your assessment that it is unlikely that RDs found in e.g., hundreds of covariates, is a legitimate RD candidate. Nevertheless, RD subgroups that are defined by a handful of covariates may still be missed if only the running variable is considered in isolation from the other pre-treatment variables, and it is these scenarios that our method fits well.
>
> **References**
>
> [1] Araneta et al. 2015: Optimum BMI Cut Points to Screen Asian Americans for Type 2 Diabetes. Diabetes Care.
>
> [2] American Diabetes Association 2015. Standards of Medical Care in Diabetes—2015: Summary of Revisions. Diabetes Care.
>
> [3] National Council of Asian Pacific Islander Physicians 2015. Screen at 23. http://ncapip.org/diabetes/screenat23/index.html
>
> [4] Asian American Diabetes Initiative 2015. Screen at 23 Campaign. https://aadi.joslin.org/en/screen-at-23
>
> [5] Kane 2003. A Quasi-Experimental Estimate of the Impact of Financial Aid on College-Going.
>
> [6] Scott et al. 2022. Regression discontinuity analysis for pharmacovigilance: statin example reflected trial findings showing little evidence of harm.

---

> ### Author Response · Authors · 2023-09-01
> **Response to Reviewer Kxby part 3**
>
> **Question 3**
>
> We did not apply the LoRD3 post-hoc validation tests in our experiments to make the comparisons as fair as possible. The LoRD3 source code was sufficiently modular such that we omitted their `validate_subsets` function from our experimental runs.
>
> **Question 4**
>
> You raise a good point on the potential interaction between bunching tests and our method. Under standard RD assumptions of continuity in covariates, partial manipulation of the running variable will not cause issues with identification, and further conditioning on any observed covariates via subgroup partitioning should not introduce them. In the case of complete manipulation, let’s suppose a scenario in diabetes treatment where wealthier patients somehow are able to change their lab results to be rounded up to the threshold. If the subgroup of wealthier patients is identified, the bunching of the running variable should become more prominent upon examination, and would not be able to pass the manipulation test. In this case, we could actually view the identification of such an invalid RD as an advantage as it would inform the practitioners of which subgroups should not be analyzed due to potential manipulation. In the other case you raise where a subgroup could potentially falsely pass a bunching test, while we don’t immediately see situations where this could be the case in our discovery method, it is an interesting question that warrants future research. Many of the CATE methods that identify local neighborhoods of data points for inference such as the honest trees we use in our approach could produce clusters that interact with post-hoc tests of continuity, and so investigating how to perform valid inference in such a sequence of model-fitting and then testing would be worthwhile.
>
> Regardless, from a practitioner’s perspective we agree that the McCrary test for manipulation should still be performed, and it is straightforward to apply post hoc tests of RD study validity like the McCrary test as an additional filtering step to our method. We elaborate on this point in a revised Section 7.
>
> **Question 5**
>
> You are correct in that our experiments do not test how our methods will perform in the setting with no subgroups, and it is possible that LoRD3 or other methods may outperform RDSGD in these situations. However, given our research question examining variation in compliance rates according to pre-treatment covariates, we specifically focus on the subgroup case in our simulations. Lord3 has a somewhat different utility by focusing on the local identification of cutpoints, and they should be seen as complementary methods; in our original submission we discuss that Lord3 can actually be integrated into Step 1 of our method to identify cutpoint candidates. At the same time, future work could evaluate how our subgroup method would perform in different data distribution settings e.g., if there are subgroups of imbalanced sizes that do not necessarily produce higher effective sample sizes.
>
> **Question 6**
>
> We want to confirm that this comment is with respect to the use of a single medical claims dataset, as we have three separate case studies (diabetes, colon cancer, breast cancer) within the claims database. We acknowledge that utilizing only a single real-world data source is somewhat limiting, but we do want to highlight that this dataset is large and has a wide scope in both time and clinical contexts, as it captures the billing events of tens of millions of patients across the United States from 2001 to 2018. We intended the scope of this work to focus on developing the methodology; as we additionally discuss the limitations of using claims data in Section 7.2, we are also planning future work that focuses on clinical practitioner insights that utilizes a more granular EHR dataset from a regional midwestern hospital system.

---

### Author Response · Authors · 2023-09-01
**Overall Review Response and Summary of Revisions**

We thank all of the reviewers for their positive and detailed feedback which have helped us improve our work. We have addressed individual questions as direct comments to each reviewer, and summarize the revisions we have made to our submission below, which are marked in blue in the updated pdf.

1. Per Reviewer 14oM’s recommendation, we have modified our algorithm to include sample splitting, and have updated the experimental results accordingly – we find that our experiments lead to the same conclusions regarding the performance of our method over baselines in the sample splitting scenario as they did in our initial submission.
2. Per Reviewer ifu5’s feedback, we add further discussion of related work in the CATE and RD literature in Section 2, include a more comprehensive introduction to RDs in Section 3.1, and incorporate explanations building off the breast cancer screening running example in Sections 3 and 5.
3. We elaborate on the practitioner use cases and power implications in Section 7 of discovered subgroups from discussion points raised by Reviewer Kxby.

---

### Decision · Action_Editors · 2023-09-21

**Recommendation:** Accept with minor revision

**Comment:**

The three reviewers recommended Accept or Learning Accept. There are however two suggestions to improve the manuscript before it is published, particularly:

1. The manuscript promotes a “bottom-up” data-driven approach to streamline and scale RD study discovery, unlocking opportunities for more interpretable and well-powered causal studies, this statement should be followed by a caution with respect to the power implications of data-driven design discovery since the bottom-up approach might reduce the subsequent power of any studies that use the design.

2. The manuscript should be specific about where and how it views the limits of the utility of multi-dimensional RD discovery. The current manuscript is certainly useful for discovering 1-3 dimensional RDs (case studies with the medical claim dataset discovers  "simple" subgroups with only 1 dimension), however more complex RDs might neither exist nor have econometric value.

**Audience:**

The audience reading TMLR with treatment effect literature in mind may find this work interesting. The framework for subset discovery should inspire future research.

All three reviewers agreed the answer to this question is yes.

**Claims And Evidence:**

The manuscript focuses on regression discontinuity (RD), a specific quasi-experimental method for evaluating causal effects from observational data, where a threshold in an observed continuous running variable determines treatment assignment. Finding the thresholds and determining to whom they apply is an important problem currently solved manually by domain experts.

The manuscript introduces Regression Discontinuity SubGroup Discovery (RDSGD), a data-driven method that identifies statistically powerful and interpretable subgroups for RD thresholds.

RDSGD was assessed in synthetic experiments and three case studies (diabetes, colon cancer, breast cancer) using a medical claims dataset consisting of over 60 million patients.

All three reviewers agreed the answer to this question is yes.